# Groundwater discharge as a driver of methane emissions from Arctic lakes

Carolina Olid 1,2,3✉, Valentí Rodellas 4, Gerard Rocher-Ros 1,2, Jordi Garcia-Orellana 4,5, Marc Diego-Feliu4,5,6,7, Aaron Alorda-Kleinglass 4, David Bastviken 8 & Jan Karlsson 1

Lateral $CH_4$ inputs to Arctic lakes through groundwater discharge could be substantial and constitute an important pathway that links $CH_4$ production in thawing permafrost to atmospheric emissions via lakes. Yet, groundwater $CH_4$ inputs and associated drivers are hitherto poorly constrained because their dynamics and spatial variability are largely unknown. Here, we unravel the important role and drivers of groundwater discharge for $CH_4$ emissions from Arctic lakes. Spatial patterns across lakes suggest groundwater inflows are primarily related to lake depth and wetland cover. Groundwater $CH_4$ inputs to lakes are higher in summer than in autumn and are influenced by hydrological (groundwater recharge) and biological drivers ($CH_4$ production). This information on the spatial and temporal patterns on groundwater discharge at high northern latitudes is critical for predicting lake $CH_4$ emissions in the warming Arctic, as rising temperatures, increasing precipitation, and permafrost thawing may further exacerbate groundwater $CH_4$ inputs to lakes.

[1] Climate Impacts Research Centre, Department of Ecology and Environmental Science, Umeå University, 90187 Umeå, Sweden. [2] Department of Forest Ecology and Management, Swedish University of Agricultural Sciences, 90183 Umeå, Sweden. [3] UB-Geomodels Research Institute, Departament de Dinàmica de la Terra i l'Oceà, Facultat de Ciències de la Terra, Universitat de Barcelona, 08028 Barcelona, Spain. [4] Institut de Ciència i Tecnologia Ambientals, Universitat Autònoma de Barcelona, 08193 Bellaterra, Spain. [5] Departament de Física, Universitat Autònoma de Barcelona, 08193 Bellaterra, Spain. [6] Department of Civil and Environmental Engineering, Universitat Politècnica de Catalunya, 08034 Barcelona, Spain. [7] Associated Unit: Hydrogeology Group, UPC-CSIC, 08034 Barcelona, Spain. [8] Department of Thematic Studies—Environmental Change, Linköping University, 58183 Linköping, Sweden. ✉email: carolina.olid@ub.edu

A major challenge to forecast future climate is constraining and regulating fluxes of greenhouse gases (GHG) such as methane ($CH_4$)[1]. $CH_4$ is a potent GHG responsible for one-quarter of the radiative forcing by all long-lived GHGs[2]. Arctic lakes represent a large and climate-sensitive natural source of $CH_4$ to the atmosphere, with emissions comparable to those from northern high-latitude wetlands[3]. In the context of climate warming, $CH_4$ emissions from Arctic lakes are expected to increase 2–3 fold by the end of the twenty-first century[4,5], potentially constituting a strong positive climate feedback. Yet, the sensitivity of $CH_4$ emissions from Arctic lakes to climate change is highly uncertain because of a poor understanding of the underlying mechanisms controlling lake $CH_4$ cycling.

$CH_4$ emissions from lakes are generally regarded to be controlled by the imbalance between in-lake processes, including $CH_4$ production[6,7] and $CH_4$ oxidation[8–10]. However, high $CH_4$ concentrations in lake waters can also result from the large supply of terrestrial $CH_4$ through groundwater discharge[11–13]. $CH_4$ inputs to lakes through groundwater and resulting emissions to the atmosphere can be important in the Arctic where wetlands (hotspots of $CH_4$ production) are abundant, and water flow paths are constrained within the shallow active layer (i.e., soil layer that thaws and refreezes annually), resulting in the supply of high loads of inorganic and organic carbon (C) to surface waters[11–13]. While external inputs of terrestrial C through groundwater discharge have been identified to have a critical influence on lake C cycling[14,15], the significance of groundwater inflows for $CH_4$ emissions from lakes has rarely been addressed.

Two recent studies on single lakes demonstrated that groundwater discharge is an important source of $CH_4$ for lakes in Alaska, suggesting that groundwater $CH_4$ inputs could entirely sustain $CH_4$ evasion rates in summer[12,13]. However, since both mire $CH_4$ production and export to receiving waters are highly dependent on environmental factors such as temperature, water table depth, active layer thickness, and topographic features[16,17], snapshot observations in single lakes or seasons may not fully represent the relevance of large-scale or year-round $CH_4$ inputs via groundwater. Extended studies covering multiple lakes and seasonal variability are thus needed to better assess the role of groundwater discharge on lake $CH_4$ emissions under different environmental conditions, not least in the Arctic, which experiences strong seasonality in terrestrial $CH_4$ dynamics, precipitation, and runoff. To fill this gap, we combined measurements of $CH_4$ and radon ($^{222}Rn$), a natural tracer of groundwater, in 10 lakes and adjacent groundwater in the Arctic region of Sweden (Fig. 1) to provide regional estimates of rates, patterns, and drivers of groundwater $CH_4$ inputs during the ice-free season.

## Results

**Groundwater $CH_4$ inputs into lakes**. We found a consistent enrichment of $CH_4$ and $^{222}Rn$ in groundwater relative to surface waters. Concentrations of $CH_4$ in groundwater (median 150 μM, interquartile range (IQR, 25th and 75th percentiles) 49–210 μM) were more than two orders of magnitude higher than in lake waters (median 0.19 μM, IQR 0.02–0.48 μM) and inlet streams (median 0.02 μM, IQR 0.02–0.37 μM) (Supplementary Fig. 1). Concentrations of $CH_4$ in groundwater are similar to those in nearby sedge and *Sphagnum* permafrost mires (13–160 μM)[18], and in the active layer from the continuous permafrost zone in Alaska (Toolik Lake: 0.63–150 μM[13]; Landing Lake: 7.8–610 μM[12]). The high $CH_4$ concentrations in groundwater suggests that even relatively low groundwater inflows into lakes may disproportionately affect lake $CH_4$ budgets. There was no evidence that groundwater $CH_4$ concentration differed between seasons (120 μM and IQR 18–240 μM in summer, 150 μM and IQR 88–210 μM in autumn)

(ANOVA, df = 1, F = 0.50, p = 0.48). Similar to $CH_4$, $^{222}Rn$ concentrations in groundwater (median 3500 Bq m$^{-3}$, IQR 2100–8800 Bq m$^{-3}$) were at least an order of magnitude higher than in lake waters (110 Bq m$^{-3}$, IQR 78–160 Bq m$^{-3}$) and inlet streams (520 Bq m$^{-3}$, IQR 260–2100 Bq m$^{-3}$) (Supplementary Fig. 2), suggesting that $^{222}Rn$ can be used as tracer for quantifying groundwater inflows into the lakes, as previously done in two studies in Alaska[12,13]. Similar ranges in $^{222}Rn$ concentration were observed in summer (3700 Bq m$^{-3}$, IQR 2600–8300 Bq m$^{-3}$) and autumn (3200 Bq m$^{-3}$, IQR 2000–8200 Bq m$^{-3}$) (ANOVA, df = 1, F = 0.087, p = 0.77).

The $^{222}Rn$ mass balance (Supplementary Fig. 3) revealed that groundwater discharge was an important water source for the lakes (Fig. 2), except for the two shallowest lakes (BD09 and BD12). Groundwater inflows varied between lakes, with median rates ranging from 0.18 to 6.4 cm d$^{-1}$. Groundwater inflows were within the range of the water discharge through the inlet streams (0.69 cm d$^{-1}$, IQR 0.20–3.0 cm d$^{-1}$, normalizing the point-source stream discharge by lake area) and comparable to those found in two other lakes in Alaska using a similar approach (0.6–2.1 cm d$^{-1}$)[12,13]. Groundwater inflows were higher in summer (range of 1.6–6.4 cm d$^{-1}$) compared to autumn (range of 0.18–3.4 cm d$^{-1}$).

$CH_4$ inputs supplied by groundwater into the study lakes ranged from 28 to 120 mg $CH_4$ m$^{-2}$ d$^{-1}$ in summer and 2.0 to 59 mg $CH_4$ m$^{-2}$ d$^{-1}$ in autumn, exceeding up to one order of magnitude the $CH_4$ inputs through the inlet streams (range of <0.01–1.3 mg $CH_4$ m$^{-2}$ d$^{-1}$) (Fig. 3a). Similar summer groundwater $CH_4$ inputs were found in Landing Lake in Alaska (32–128 mg $CH_4$ m$^{-2}$ d$^{-1}$)[12]. Lower $CH_4$ inputs through groundwater were found in Toolik Lake in summer (1.6–11 mg $CH_4$ m$^{-2}$ d$^{-1}$)[13], likely due to the lower $CH_4$ concentrations in groundwater from the active layer (8–35 μM) compared to lakes in this study (50–210 μM).

To better understand the significance of groundwater discharge for lake $CH_4$ cycling, we compared groundwater $CH_4$ inputs with total $CH_4$ emissions from the lakes. Diffusive $CH_4$ fluxes to the atmosphere in summer ranged from 0.70 to 7.6 mg $CH_4$ m$^{-2}$ d$^{-1}$. Lower diffusive $CH_4$ fluxes were observed in autumn, ranging from <0.01 to 2.3 mg $CH_4$ m$^{-2}$ d$^{-1}$. As ebullition fluxes were not directly measured here, we used the results from 9 years of flux measurements in three lakes within the Stordalen mire, close to the study lakes (Fig. 1), to estimate the potential contribution of ebullition in our lakes[19]. For lakes in Stordalen, diffusive $CH_4$ emission accounted for 17–52% of the ice-free $CH_4$ flux, with the remainder being emitted via ebullition. The Stordalen lakes are situated in a unique palsa mire complex and are not fully representative of this landscape, likely leading to ebullition contribution more to the total lake emissions than other lakes in the region. Hence, to estimate maximum ebullition and maximum total lake $CH_4$ emission—thereby making our comparisons between groundwater $CH_4$ inputs and total emissions conservative—we assumed that diffusive flux and ebullition in all lakes accounted for 17 and 83% of the total atmospheric $CH_4$ emission, respectively. Thus, potential ebullition rates were estimated to range from < 0.1 to 37 mg $CH_4$ m$^{-2}$ d$^{-1}$. This range is similar to ebullition fluxes reported for small and midsize glacial and post-glacial lakes (IQR 0–15 and 3–27 mg $CH_4$ m$^{-2}$ d$^{-1}$) across the boreal and Arctic region[20]. These potential ebullition rates yield maximum total $CH_4$ emissions that ranged from 4.1 to 44 mg $CH_4$ m$^{-2}$ d$^{-1}$ in summer and from <0.1 to 13 mg $CH_4$ m$^{-2}$ d$^{-1}$ in autumn. Overall, these potential total $CH_4$ emissions were of the same order of magnitude as the $CH_4$ inputs supplied by groundwater (Fig. 3a), suggesting that groundwater $CH_4$ discharge can match total lake $CH_4$ emissions at a regional scale.

Our results show that groundwater discharge to Arctic lakes is a pervasive external source of $CH_4$, with the potential to sustain

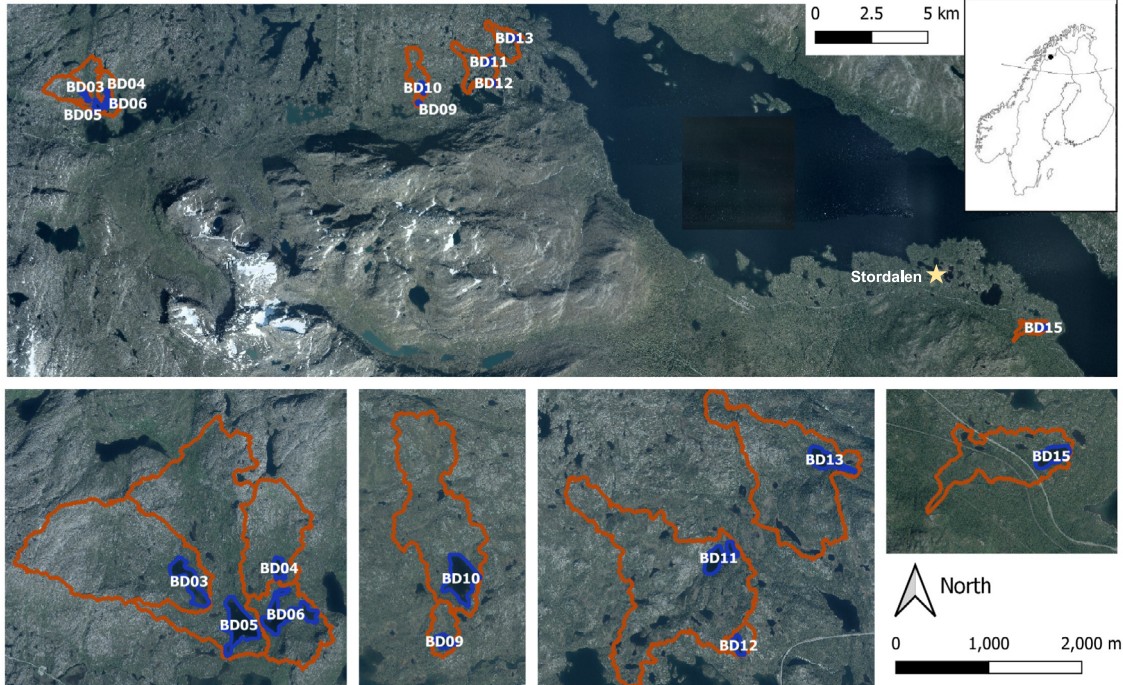

**Fig. 1 Map of the sampled lakes in Northern Sweden.** Blue color indicates the study lakes and red lines show the corresponding catchments. The yellow star indicates the location of the Stordalen mire (Image source: © Lantmäteriet, 2021).

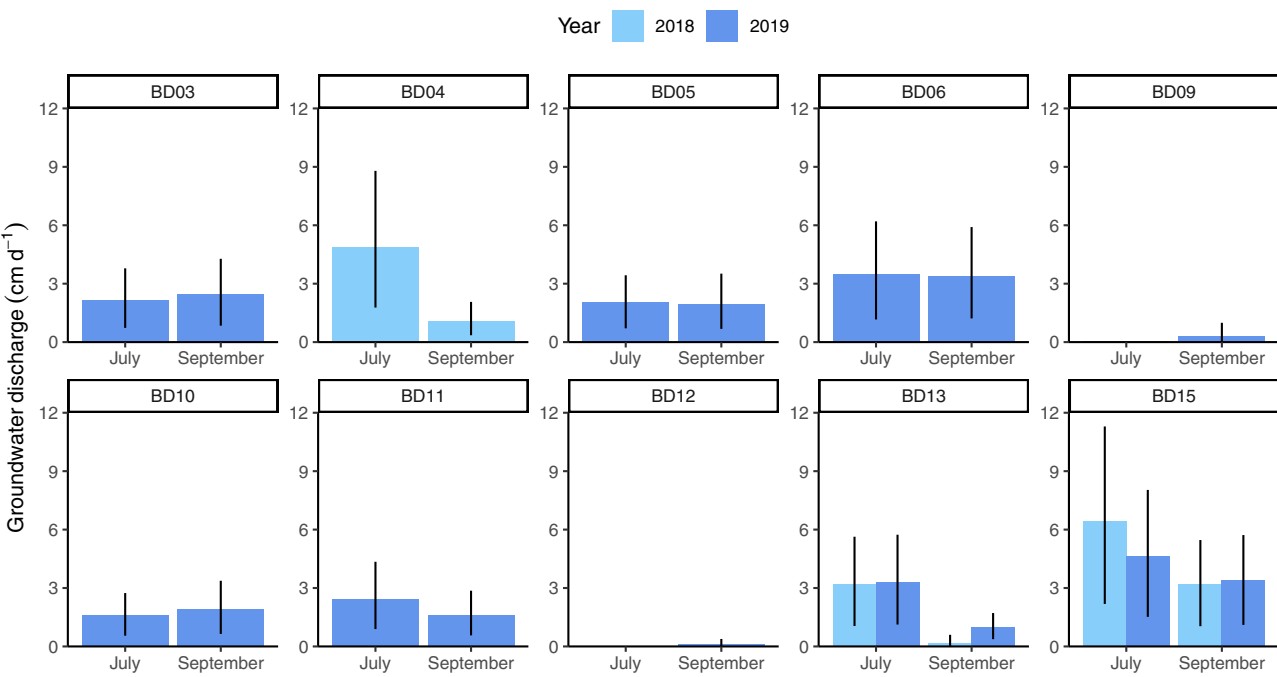

**Fig. 2 $^{222}$Rn-derived groundwater inflows into the study lakes.** Values and error bars are obtained from the median and the 25th and 75th percentiles, respectively, of the 1000 Monte Carlo simulations for each lake and season.

total lake $CH_4$ emissions. However, other lake processes that can substantially control lake $CH_4$ emissions need to be considered. For instance, large amounts of $CH_4$ are produced in sediments and transferred to lake waters, and a large share of the lake $CH_4$ is consumed via oxidation in the water column[8–10]. Nevertheless, $CH_4$ production[21–24] and oxidation rates[9,21–23,25] in lakes across the Arctic are comparable in magnitude to groundwater $CH_4$ inputs found in this study (Fig. 4), which emphasizes the relevance of groundwater discharge as an important mechanism

controlling lake $CH_4$ budgets. These results help to understand the disproportionate role of Arctic lakes as a landscape source of $CH_4$ and highlight the need to consider groundwater $CH_4$ inputs to understand lake $CH_4$ emissions at the catchment level. The importance of groundwater inflows on controlling lake $CH_4$ emissions is further supported by the positive correlation ($R^2 = 0.61$, $p = 0.005$) between groundwater discharge rates and total $CH_4$ emissions during summer (Fig. 3b). Unlike previous investigations based on single lakes[12,13], this study reveals that

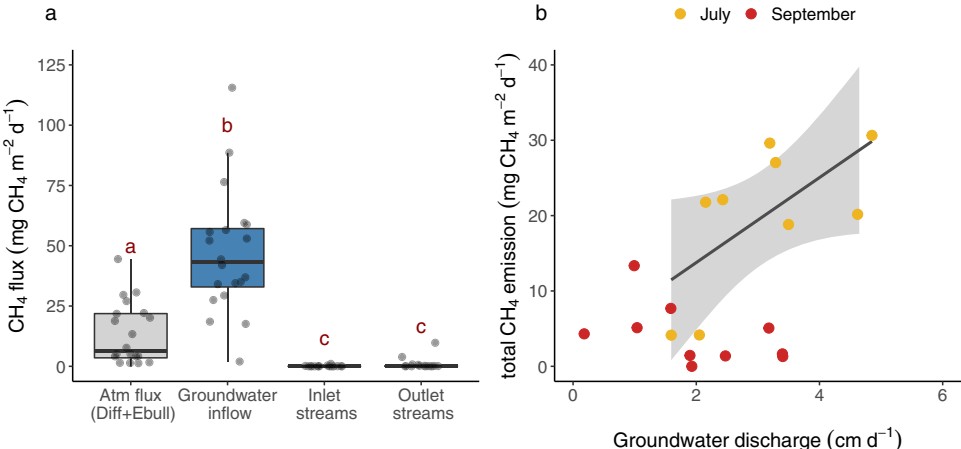

**Fig. 3 CH₄ fluxes from different sources and relationship between total CH₄ emissions and groundwater inflow rates. a** Inputs of CH₄ to the study lakes are groundwater inflow and inlet streams, while outputs are emissions to the atmosphere (Atm flux) (by diffusion (Diff) and ebullition (Ebull)) and outlet streams. Box plots for groundwater CH₄ inputs were generated by considering median values reported at each lake for each season. The boundaries of each box plot indicate the 25th and 75th percentiles of these fluxes, and the solid line in each box marks the median. Different lower-case letters indicate differences between water sources. The data used to generate the box plots is represented with gray circles. **b** In summer, as groundwater inflow rates increase, higher atmospheric CH₄ emissions are found. The solid line represents the linear regression between total CH₄ emissions and groundwater inflow rates ($y = (6.5 \pm 1.7)\, x + (0.1 \pm 6.2)$, $df = 8$, $F = 15$, $p < 0.005$). The shaded area represents 95% confidence intervals.

groundwater discharge is a key process controlling lake CH₄ emissions at high latitudes and represents an important source of CH₄ at a regional scale. Therefore, including groundwater discharge to lakes in the global CH₄ cycling may improve climate predictions[26].

**Spatial and temporal patterns of groundwater CH₄ inputs to lakes.** There was a large variation in the magnitude of groundwater inflows among lakes and seasons. The partial least squares regression (PLS) showed that selected catchment (percentage of wet zones and open mires, catchment area, and slope) and lake characteristics (mean depth) together with precipitation explained 72% of the variability in groundwater inflows among lakes (Supplementary Fig. 4). The best multiple linear regression model showed that the best predictors of groundwater inflows were mean lake depth (depth), wet area coverage (wetzone) in the catchment, and catchment slope (slope), with the resulting model explaining 45% of the variance ($Q_{gw} = 7.3 + 3.5 \cdot \text{depth} - 5.7 \cdot \log_{10}(\text{wetzone}) - 0.27 \cdot \log_{10}(\text{slope})$; $df = 20$, $F = 7.2$, $p < 0.002$). Groundwater inflows were positively related to lake depth, likely reflecting the higher water interception and groundwater connectivity of large and deep lakes compared to shallow systems. Contrary to our expectations, the strongest explanatory variables, wet area, and mire catchment, showed a negative relationship with groundwater inflows. We initially expected that mire cover would positively affect groundwater inflows to lakes as the cover of wet areas would represent the hydrological connectivity of lakes with the catchment. A possible explanation for this result is that increasing mire cover is related to flatter catchments, reducing the hydrological gradient and, consequently, reducing the lateral water transport and increasing the groundwater residence time in peatlands[27]. Our findings highlight the need to consider how groundwater inflows overlay landscape patterns of CH₄ production to better assess lake CH₄ emissions, as they may contrast and lead to unexpected responses on terrestrial CH₄ export through groundwater. The complexity of hydrological pathways in a catchment and the number of variables involved in groundwater transport (e.g., permafrost coverage, preferential flow paths, hydraulic conductivity, and geological heterogeneities)[27–30] prevents any further assessment of the spatial variance of groundwater inflows. Regardless, we show that it is possible to predict

groundwater CH₄ inputs into lakes based on spatial variables, which opens the door to future inclusions in regional assessments and/or Earth system models.

The impact of groundwater discharge on lake CH₄ emissions varied seasonally, as shown by the strong positive correlation between groundwater inflow rates and CH₄ evasion in summer that was not observed in autumn (Fig. 3b). This seasonal divergence indicates that the mechanisms driving groundwater CH₄ inputs to lakes may be sensitive to climatic conditions and likely reflect seasonal changes in hydrological and biological drivers. For instance, snow melt during spring typically increases groundwater recharge compared to the frozen period[31–33], consistent with seasonal water table level variations in surficial aquifer piezometers in the Arctic[34,35]. The lower magnitude of groundwater inflows in autumn compared to summer observed in this study agrees with the general hydrological cycle in the region[36], with higher water flows in early summer compared to the autumn (Supplementary Fig. 5).

Furthermore, changes in biological processes may also affect the CH₄ pool available for export via groundwater discharge. Production rates of CH₄ in mires are expected to be higher in summer compared to autumn due to warmer soil temperatures[37,38]. However, CH₄ concentration in mire groundwater measured in this study did not differ between seasons, possibly because of the lower groundwater discharge (i.e., longer water residence time in the soil) in combination with lower CH₄ production rates in autumn, which could result in similar levels of accumulated CH₄. Overall, the strong correlation between groundwater inflows and atmospheric CH₄ evasion likely results from a higher discharge in early summer, although the role of biological processes in CH₄ cycling needs further attention[8,39]. The seasonal variability in groundwater inflows into lakes observed here emphasizes the need for a better characterization of temporal variations of this external driver for its inclusion in broad-scale estimates of lake CH₄ emissions.

**Implications for Arctic lakes under climate change.** This study unraveled the important role and drivers of groundwater discharge for CH₄ emissions from Arctic lakes. The comparison of the magnitudes of groundwater CH₄ inputs to lakes with other

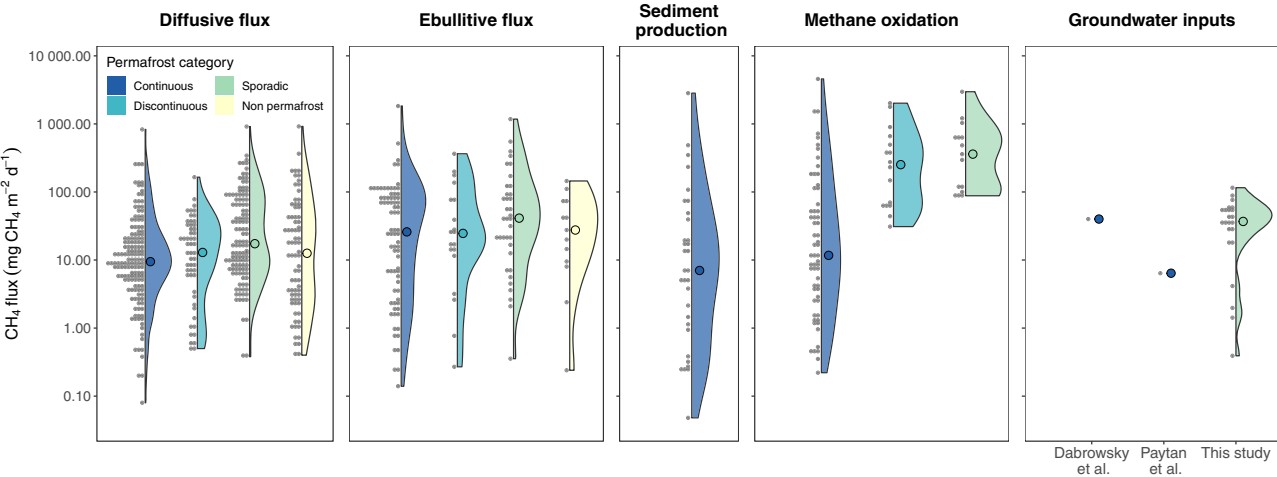

**Fig. 4 Comparison of groundwater CH$_4$ inputs with other CH$_4$ fluxes in Arctic lakes.** Atmospheric fluxes (diffusion and ebullition) were extracted from the Boreal-Arctic Wetland and Lake Methane Dataset (BAWLD-CH4)[20]. Sediment production[21-23] and oxidation[9,21,23] rates were obtained from incubation experiments found in the literature (see Supplementary Data 1).

potential CH$_4$ sources and sinks (Fig. 4) suggests that groundwater discharge is a major mechanism controlling CH$_4$ cycling in Arctic lakes. Spatial variability in rates of groundwater inflow is mainly derived from the physical-hydrological characteristics of the catchment-lake continuum (lake morphology and vegetation cover). Seasonally, groundwater CH$_4$ inputs to lakes are influenced by hydrological (groundwater recharge) and biological drivers (CH$_4$ production rates). This information on the spatial and temporal patterns of groundwater discharge in permafrost environments is fundamental to understanding the dynamics of landscape-level lake CH$_4$ emissions and associated responses to climate warming.

Our results indicate that the multiple facets of climate change in the Arctic may exacerbate the magnitude of groundwater CH$_4$ inputs to lakes and subsequent emissions to the atmosphere. For instance, rising temperatures result in extensive degradation of permafrost[40,41], deepening active layer thickness, and higher amounts of organic C exposed to decomposition, which can fuel methanogenesis in anoxic environments[42]. At the same time, CH$_4$ production is highly dependent on temperature[43] and thus may also increase with current warming, increasing CH$_4$ concentration in groundwater from the active layer. Furthermore, precipitation is expected to increase up to 40% in the Arctic by the end of this century[44] which, together with enhanced groundwater recharge due to permafrost thaw and tundra greening[45], may reshape the hydrology of the Arctic and increase groundwater inflows into lakes. Taken together, all these long-term changes can potentially increase the magnitude of lake CH$_4$ emissions derived from groundwater inputs. Thus, groundwater CH$_4$ inputs constitute an often-overlooked feedback in the ongoing climate change that needs to be recognized and addressed appropriately to better forecast future CH$_4$ emissions from Arctic lakes.

## Methods

**Study site.** The Torneträsk catchment in Arctic Sweden (68.40°N 18.90°E; Fig. 1) lies within the discontinuous permafrost zone along the 0 °C isotherm[46]. Mountain permafrost is found approximately above 880 m a.s.l., whereas at lower elevations permafrost is present on north-facing slopes and wind-exposed areas because of the lack of an insulating snow cover during winter[47] (Supplementary Fig. 6). The area is characterized by the transition between tundra and nordic mountain birch forest[48-50] (Supplementary Fig. 7). The historical mean annual temperature for the study lakes for the 1970–2000 period ranged between −0.1 to −1.4 °C[51]. During the past decade, however, the temperature has risen above 0 °C[52]. The area exhibits a strong climatic gradient, with a general decrease in regional precipitation and annual temperature amplitudes eastwards. The highest precipitation (~1000 mm yr⁻¹) occurs near the Norwegian border, while the lowest (~300 mm yr⁻¹) is around Abisko.

About 15% of the total catchment area is composed of lakes, concentrated in low elevation areas where this study was performed[53]. We selected lakes ($n = 10$) across the precipitation gradient (Fig. 1) to investigate seasonal patterns on CH$_4$ inputs supplied by groundwater to Arctic lakes. All selected lakes were small (area between 1.8 and 11.6 ha), with an average mean depth of 2.9 m and water volumes ranging from 24,000 to 760,000 m³.

**Estimating groundwater CH$_4$ inputs into lakes using ²²²Rn.** We quantified groundwater inflow rates into the study lakes using the noble gas radon (²²²Rn) as tracer of groundwater inputs. ²²²Rn ($T_{1/2}$ = 3.82 d) is a radioactive isotope produced in the uranium (²³⁸U) decay series. Owing to its high enrichment in groundwater and its conservative behavior in waters[54], ²²²Rn is an ideal geochemical tracer to detect and quantify groundwater inflow rates into surface waters[55,56]. We estimated groundwater inflow rates to the study lakes using a ²²²Rn mass-balance approach[57,58]. Major model assumptions include steady-state conditions over a relatively short period (1–3 days, comparable to ²²²Rn residence time in the system[59]) and a well-mixed water column.

Our survey showed that the ²²²Rn signal was relatively uniformly distributed horizontally and vertically in all the lakes, indicating well-mixed water columns. The mass-balance approach is based on accurately constraining all the ²²²Rn sources (groundwater inflow ($F_{gw}$), diffusion of ²²²Rn from bottom sediments ($F_{diff}$), discharge from the inlet streams ($F_{inlet}$), and in situ ²²²Rn production from decaying ²²⁶Ra dissolved in the water column ($F_{Ra}$)), and sinks (evasion to the atmosphere ($F_{atm}$), losses through the outlet streams ($F_{outlet}$), and radioactive decay ($F_{decay}$)). The change in ²²²Rn content over time [Bq d⁻¹] can thus be described as:

$$\frac{\partial(C_{Rn,lake}V)}{\partial t} = F_{gw} + F_{diff} + F_{inlet} + F_{Ra} - F_{atm} - F_{outlet} - F_{decay} \quad (1)$$

This equation can also be expressed as:

$$\frac{\partial(C_{Rn,lake}V)}{\partial t} = Q_{gw}C_{Rn,gw} + f_{diff}A + Q_{inlet}C_{Rn,inlet} + \lambda V C_{Ra,lake} - f_{atm}A - Q_{outlet}C_{Rn,outlet} - \lambda V C_{Rn,lake} \quad (2)$$

where $Q_{gw}$ [m³ d⁻¹] is the advective groundwater inflow; $C_{Rn,gw}$, $C_{Rn,lake}$, $C_{Rn,inlet}$, and $C_{Rn,outlet}$ [Bq m⁻³] are the ²²²Rn concentrations in groundwater, lake water, inlet and outlet streams, respectively; $Q_{inlet}$ and $Q_{outlet}$ are the mean water flow rates [m³ d⁻¹] for the inlet and outlet streams, respectively; $C_{Ra,lake}$ [Bq m⁻³] is the ²²⁶Ra concentration [Bq m⁻³] in lake water; $f_{diff}$ is the molecular diffusion flux of ²²²Rn from underlying sediments [Bq m⁻² d⁻¹]; $f_{atm}$ is the atmospheric flux of ²²²Rn to the atmosphere [Bq m⁻² d⁻¹], $\lambda$ is the ²²²Rn decay constant [d⁻¹]; and $A$ [m²] and $V$ [m³] are the lake surface and volume, respectively. Note that this approach assumes that all lakes are not losing water via groundwater, resulting in estimates of the minimum amount of groundwater discharging into the lakes.

We assume that the ²²²Rn concentration in lake water is nearly in steady-state $\left(\frac{\partial(C_{Rn,lake}V)}{\partial t} = 0\right)$ during the residence time of ²²²Rn in the water column. The ²²²Rn residence time ($\tau$, [d]) in the lakes was estimated following the equation[59]:

$$\tau = \frac{1}{\lambda + \frac{Q_{outlet}}{V} + \frac{k}{h}} \quad (3)$$

where $h$ [m] and $k$ [m d⁻¹] are the mean depth of the lake and the gas transfer velocity for ²²²Rn (see below, section "Atmospheric fluxes"), respectively. The rest of the parameters are described in Eq. 2. Considering an average $k$ and $h$ of

0.67 m d$^{-1}$ and 2.8 m, respectively, and using the average $Q_{outlet}/V$ measured in the study lakes (0.06 d$^{-1}$), the average residence time of $^{222}$Rn for the study lakes is ~2–3 days. Steady-state conditions over three days are thus a reasonable assumption considering the relatively stable environmental conditions in the days before the sampling (e.g., no precipitation events, minor changes in wind regimes and temperatures).

The mass balance in Eq. 2 was used to estimate the flux of $^{222}$Rn supplied by groundwater ($F_{gw} = Q_{gw}C_{Rn,gw}$) into each lake under steady-state conditions. Uncertainties associated with $F_{gw}$ were deterministically estimated by propagating the uncertainties of the individual terms in Eq. 2 (see Supplementary Methods).

The estimated $^{222}$Rn flux ($F_{gw}$) and its uncertainty were then used for quantifying groundwater inflows and associated CH$_4$ inputs to each lake based on a Monte Carlo analysis. The analysis consisted of generating 1000 values of $F_{gw}$ for each lake (following a normal distribution based on calculated $F_{gw}$ and its uncertainty). Each generated $F_{gw}$ was then divided by a $^{222}$Rn concentration in groundwater ($C_{Rn,gw}$) to derive a groundwater flow, and by a $^{222}$Rn to CH$_4$ concentration ratio in groundwater (i.e., $C_{Rn,gw}/C_{CH4,gw}$), for calculating CH$_4$ inputs. Both $C_{Rn,gw}$ and $C_{Rn,gw}/C_{CH4,gw}$ were randomly selected from all groundwater samples collected ($n = 41$), producing a 1000-length list of groundwater flows and groundwater CH$_4$ fluxes for each lake. Final fluxes are reported as the range of the median values for all the lakes. Groundwater inflows and associated CH$_4$ inputs are reported in the manuscript as the median and the 25th and 75th percentiles of the 1000 simulations for each lake and season.

**Sampling and analyses.** The first survey in 2018 included three lakes (BD04, BD13, BD15) sampled in June (21–25) and September (8–12). In July (27–28) 2018, a short sampling campaign was conducted to collect groundwater samples from the active layer. For the second survey in 2019, nine lakes (BD03, BD04, BD05, BD06, BD09, BD11, BD12, BD13, BD15) were sampled in July (9–13) and September (13–20). Dates were selected to capture summer high flow and autumn base flow conditions.

*Surface water.* Surface lake water samples were collected from the shore and open-water areas of the lakes using a submersible pump (Supplementary Fig. 8). A deep (4 m depth) water sample was collected from the deepest lake point. Water samples were also collected from the main inlet and outlet streams.

For $^{222}$Rn analyses, water samples were filled into 1.5 L polyethylene terephthalate (PET) bottles, minimizing water-air contact to prevent $^{222}$Rn degassing. Shortly after collection, $^{222}$Rn activities were determined using a Durridge RAD7 electronic radon-in-air monitor coupled to the RAD7 Soda bottle aerator kit accessory. $^{222}$Rn measurements were decay-corrected and converted to water concentrations using the air-water partitioning of $^{222}$Rn corrected for water salinity and temperature[60].

The concentration of $^{222}$Rn supported by $^{226}$Ra decay in the lake water column was determined by measurements of the $^{226}$Ra concentration in lake water from five of the study lakes. Large volumes (30–50 L) of lake water were collected using a submergible pump and filtered slowly (<1.0 L min$^{-1}$) through a column loosely filled with MnO$_2$-impregnated acrylic fiber (ca. 25 g dry) to quantitatively extract Ra isotopes[61,62]. Fibers were rinsed with Milly-Q water, incinerated (820 °C, 16 h), ground, and transferred to hermetically sealed counting vials. Samples were analyzed using a well-type Ge detector (Canberra model GSW120) after storing the samples for a minimum of three weeks to ensure the radioactive equilibrium between $^{226}$Ra and its daughters.

Dissolved CH$_4$ concentrations were determined by analyzing the headspace of gas-tight vials (22 mL vials, PerkinElmer Inc., U.S.) after addition of 20 μL of 4% HCl to 4 mL sampled water, using a gas chromatograph (Clarus 500, PerkinElmer Inc., USA). A gas mixture with known concentrations of CH$_4$ (10 and 500 ppm) was prepared, stored, and analyzed as standards together with each batch of samples. Triplicate analyses of the standards were within 2% coefficient of variation. In a few samples ($n = 4$), CH$_4$ concentrations were below atmospheric saturation and outside the detection limits of the instrument. Those values were assumed to be in equilibrium with the atmosphere.

*Groundwater.* Groundwater (20–40 cm deep) samples ($n = 41$) were collected from mire areas right at the lake shoreline (Supplementary Fig. 9) using a direct-push well-point piezometer coupled to a gas-tight syringe and tubing, minimizing the water-air contact[57]. For $^{222}$Rn analysis, 10 mL of filtered (0.45 μm) groundwater were collected and directly transferred to 20 mL polyethylene vials prefilled with a 10 mL high-efficiency liquid scintillator cocktail[63]. Concentrations of $^{222}$Rn were analyzed using an ultra-low-level liquid scintillation counter (Quantulus 1220) with alpha-beta discrimination counting (background of 0.02–0.07 cpm; efficiency of 1.5–3.0, depending on the quenching factor of the sample). Samples for dissolved CH$_4$ were collected simultaneously following the sampling procedure described above.

*Physicochemical parameters.* Water temperature, dissolved oxygen (DO), and specific conductivity were measured in situ in lake and stream waters using a calibrated handheld water monitor (Yellow Springs Instrument ProSolo). A depth profile of temperature and DO was measured every 0.5 m from the surface at the deepest point of each lake. Water pH and conductivity were measured using pH and conductivity electrodes in the laboratory.

Discharge estimates from inlet and outlet streams were measured using an electrode magnetic flow meter (model 801 EC Meter; Valeport, Devon, U.K.) (in 2018) and based on salt slug injections[64] (in 2019). Wind-speed, rainfall, air temperature, and air pressure data were acquired from weather stations permanently installed at the shore close to each lake.

*Sediments.* In July 2019, lake sediment cores were collected from all the lakes (except lake BD05) using a standard sediment corer made from PVC pipes. Three sediment cores were sliced into 1 cm thick sections, weighted, and dried to calculate porosity and dry bulk density. The remaining sediment cores were reserved for laboratory sediment incubation experiments[65,66]. The incubation experiments were used to constrain the diffusive $^{222}$Rn inputs from underlying sediments ($f_{diff}$) for the $^{222}$Rn mass balance and to obtain an independent estimate of the $^{222}$Rn concentration in the groundwater end-member ($C_{Rn,gw}$) (used only for comparison with direct measurements of groundwater samples; Supplementary Fig. 10).

*Lake and catchment characteristics.* Echo sounding was done at transects 10–20 m apart for bathymetric analysis. Lake average depth and surface area were calculated using ReefMaster v2.0 and the add-on volumes and areas[67]. Catchment delineations were made from a 2-m digital evaluation model[68] using Whitebox GAT[69], allowing to burn channels through road culverts[70]. Catchment slope was calculated using the "slope" function in the "Spatial Analyst" toolset in Arcmap 10.8 (ESRI). Catchment forest and mire cover were calculated by overlying vegetation maps[68] to the catchment areas. Catchments for each lake were delimited using a flow direction and flow accumulation model for the landscape, derived from the national digital elevation model (DEM) with a horizontal resolution of 2 m (Lantmäteriet; https://www.lantmateriet.se/). This analysis was performed using the hydrological toolbox from ArcMap 10.8 (ESRI 2019 Redlands, CA: Environmental Systems Research Institute). Once catchments were delineated, several catchment properties were extracted from the DEM (elevation range, average catchment slope, average aspect). Another landscape property used was modeled soil moisture (Soil moisture map, Dept. of Forest Ecology and Management, Swedish University of Agricultural Sciences). This machine learning product represents soil wetness in a scale from 0 to 100[71], and here we used it as a proxy of surface hydrological connectivity, calculating the average value for each catchment. We characterized the fraction of the catchment above 70% wetness to quantify the catchment hydrological connectivity, which captures water-saturated zone areas (Supplementary Fig. 11). Maps were produced using QGIS (QGIS.org, 2021. QGIS Geographic Information System. QGIS Association. http://www.qgis.org), using as layers the global permafrost map[72], Swedish land cover, and national orthophotos (Lantmateriet; https://www.lantmateriet.se/).

*Atmospheric fluxes.* The flux of $^{222}$Rn and CH$_4$ to the atmosphere were calculated as:

$$f_{atm} = k_{gas}\left(C_{gas,lake} - C_{gas,air}\right) \tag{4}$$

where $k_{gas}$ [cm d$^{-1}$] is the gas transfer velocity for the corresponding gas at the measured temperature, $C_{gas,lake}$ and $C_{gas,air}$ [Bq m$^{-3}$] are the gas concentration measured in the lake and the concentration expected when the lake is in equilibrium with the atmosphere, respectively.

For $k$, we used a wind-based model developed by Klaus and Vachon[73] based on empirical $k$ estimates from 46 globally distributed lakes data (see Supplementary Methods). This model fitted the study lakes as the range of conditions in terms of wind-speed (from 0 to 16 m s$^{-1}$) and lake surface area (from 0.018 to 0.11 km$^2$) cover a substantial range of the calibration dataset (wind-speed from 0 to 13 m s$^{-1}$; lake surface area from 0.018 to 1342 km$^2$). To weigh the different influences on $^{222}$Rn budgets of degassing events depending on their proximity to the sampling time, we used a weighting factor to the hourly wind-speed data[59]. To evaluate the uncertainties associated to $k$ parametrization, two other empirical equations for $k$ estimates commonly used in lakes were used[74,75].

*Compilation of existing data of CH$_4$ fluxes from Arctic lakes.* We compiled data on CH$_4$ fluxes (groundwater inputs, diffusion, ebullition, sediment production, and oxidation) from several studies across the Arctic to evaluate the importance of groundwater discharge in lake CH$_4$ cycling. This compilation includes warm-season (May–October depending on the location) diffusive and ebullition fluxes extracted from the Boreal-Arctic Wetland and Lake Methane Dataset (BAWLD-CH4)[20] that is available at the Arctic Data Center (https://doi.org/10.18739/A2C824F9X). Sediment production[21–23] and oxidation[9,21,23] rates were obtained from incubation experiments found in the literature and include results from 46 lakes across continuous, discontinuous, sporadic, and non-permafrost regions. The compiled values and the original sources and further details are found in Supplementary Data 1 and Data 2.

*Statistics and error estimates.* Lake $^{222}$Rn and CH$_4$ concentrations were reported as the mean of collected samples (±1 standard deviation). Differences in $^{222}$Rn and CH$_4$ concentrations between water sources (groundwater, streams, and lake waters) and between seasons (summer and autumn) were tested by analysis of variance (ANOVA), followed by Tukey-Kramer HSD post hoc test to identify differences

between groups. All statistical tests were considered statistically significant at $p < 0.05$. We followed a two-step procedure to analyze spatial patterns in groundwater inflows. First, partial least squares regression (PLS) was conducted to detect correlation structures in the dataset and to rank the relative importance of catchment and lake characteristics on groundwater discharge rates. PLS is especially suitable for correlated covarying predictor variables and when there are more predictor variables than observations[76,77]. Catchment (area, slope, percentage of wetzone and peatland cover) and lake (area, depth) characteristics, and precipitation were used as predictor variables. Cross-validation was used for selecting the optimal number of components that minimize the prediction errors (RMSE). The variable importance in projection (VIP) coefficients was calculated to classify predictors according to their explanatory power of the dependent variable. Then, multiple stepwise linear regression analysis was used to identify relationships among the most relevant predictor variables and groundwater inflows. Models were selected by considering all subsets on adjusted $R^2$ and goodness-of-fit using the Akaike Information Criteria (AIC), such that the most parsimonious model yielded the lowest AIC value. AIC measures both the model fit and complexity and is used in model selection to reduce over-fitting. Predictor variables were evaluated for multicollinearity using Spearman correlation. Correlations between predictor variables with a $p$-value less than 0.05 were considered multicollinear and removed from the models. Response and predictor variables were visually inspected for normality and log-transformed to improve normality. All data processing, statistics, and figures were done in R (version 4.1.0)[78], using the PLS[79], and ggplot2[80] packages.

## Data availability
Source data can be found in the supplementary materials of this paper.

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

## Acknowledgements

We thank Karl Heuchel, Joan Manuel Bruach, Julia Papenhausen, Maike Strack, Alberto Zannella, and Mathilde Schnuriger for invaluable assistance in the field. Sven Norman and Marcus Klaus for bathymetry maps and geographical information. Danny CP Lau and Isolde Puts for their help with the PLS analyses, and the staff at the Abisko Scientific Research Station (ANS) for infrastructure, logistic and technical support. This study has been made possible by data provided by Abisko Scientific Research Station (ANS) and the Swedish Infrastructure for Ecosystem Science (SITES). This study was financially supported by FORMAS (Grant no. 2018-01217) with a grant awarded to C.O. and by the Swedish Research Council (2016-05275) and Knut and Alice Wallenberg Foundation (2016.0083) with grants awarded to J.K. D.B. was supported by the European Research Council (ERC; grant 725546), the Swedish Research Council (grant 2016-04829), and FORMAS (grant 2018-01794). J.G.O. acknowledges the financial support of the Spanish Ministry of Science, Innovation and Universities, through the "Maria de Maeztu" program for Units of Excellence (CEX2019-000940-M) and the Generalitat de Catalunya (MERS; 2017 SGR–1588). V.R. acknowledges financial support from the Beatriu de Pinós postdoctoral program of the Catalan Government (Generalitat de Catalunya) (2019-BP-00241). M.D.F. acknowledges the economic support from the FI- 2017 fellowships of the Generalitat de Catalunya autonomous government (2017FI_B_00365). A.A.K. acknowledges financial support from ICTA "Unit of Excellence" (MinECo, MDM2015-0552-17-1) and PhD fellowship, BES-2017-080740.

## Author contributions

C.O. and J.K. designed the study. C.O., V.R., J.G.-O., M.D.-F., A.A.-K. performed fieldwork and lab analysis with infrastructure support from C.O., J.K., and J.G.O. C.O. performed data analyses, and G.R.-R. helped generate figures. C.O., V.R., and M.D.-F. performed uncertainty analysis. J.K. and D.B. provided input on $CH_4$ fluxes inter-pretation. C.O. interpreted results and drafted the manuscript with main inputs from G.R.-R., V.R., and J.G.-O. C.O. wrote the final version of the manuscript with con-tribution from all co-authors.

## Funding

## Competing interests

The authors declare no competing interests.
