## [Peer Review File · Nature Communications]

Groundwater discharge as a driver of methane emissions from Arctic lakesREVIEWER COMMENTS

Reviewer #1 (Remarks to the Author):

General comments:

This is very well-written manuscript (ms) and I did not have issues neither with its writing style nor with the structure. Additionally, all the collected data used in this ms are easy to access and verify. I commend the authors for presenting their work in this clean format that was easy to follow.

The authors have set ambitious goals to shed light on the main drivers of methane (CH₄) emissions from lakes in the Arctic. Hence, the ms presents the results from (1) seasonal regional scale assessments of CH₄ fluxes derived by groundwater discharge for 10 lakes in the Arctic, and (2) comparison to concurrently measured atmospheric fluxes.

The idea that gw discharging to Arctic lakes is a source of GHG (specifically CH₄) is not new. Paytan et al. (2015) was the first work that demonstrated the magnitude of CH₄ fluxes from Lake Toolik (Alaska) using the same approaches. After revising the published research on the subject, I think that the somewhat novelty of this work is that it attempts to examine the drivers of CH₄ fluxes. This examination was done using concurrently collected environmental data and PCA analyzes (Figure 4). While these analyzes could be useful with certain degree of confidence, but I found the authors' attempt to reveal the mechanisms that drive the changes in fluxes relatively weak.

Overall, I think this is great work that could be published in some other lower impact journal. My main concerns are that the work makes some progress in the right direction of bringing more attention to gw as a source of CH₄ to Arctic lakes, yet (1) this is not a new idea, and (2) the identified drivers (e.g. temperature) are somewhat already known/established in the field of high latitude wetland studies (e.g. Saarnio et al., 1997; Luascu et al, 2012 to mention just a couple).

Does the work support the conclusions and claims, or is additional evidence needed?

Some claims are better supported than others. For example, the significance of gw as a source of CH₄ is well supported by the data after being compared to atmospheric fluxes (with one caveat, they assessed only gas diffusion and did not assess ebullition). However, their analyzes on the drivers such as temperature, water depth etc., are not well-supported to my opinion. They made some assumption in their interpretations and suggest "more future research on the subject needs to be done".

Are there any flaws in the data analysis, interpretation and conclusions? - Do these prohibit publication or require revision?

I don't find any flaws in the data analysis, but some of the interpretations and then statements are somewhat far reaching. For example, in order to demonstrate the magnitude of the gw-derived CH₄ fluxes, they compared them to atmospheric fluxes (which is great!). However, they only calculated the diffusive CH₄ fluxes to the atmosphere but neglected the ebullition (bubbles formed withing the permafrost and the lake water column). On page 7-8 line 125-129 they concluded, that "While diffusive fluxes are often lower than ebullition and often account for 20-50% of the sum of the total CH₄ in similar lakes, their results suggest that gw inflow of CH₄ is comparable in magnitude to the CH₄ flux to the atmosphere by BOTH diffusive flux and ebullition." I consider this is a relatively big assumption and find it a weakness in their interpretations.

Similar statement is made on page 8 lines 130-144. At the beginning of the paragraph the authors state that their work "confirms" that gw is an overlooked source of CH₄ (hence I understand not a new idea), but their work is unique in a way that it is based on multiple lakes, i.e. it is an ubiquitous process. While I think taking their findings just in the context of high latitude lakes is perhaps true, Paytan et al. (2015) and Lecher et al (2015) demonstrated in two-year study that this process is indeed ubiquitous not only in lakes, but also across various permafrost coverage in marine and fresh water systems. I find the findings by Paytan's group having more weight on the arguments of ubiquitous distribution. Hence, the current statement not extremely significant.

Is the methodology sound? Does the work meet the expected standards in your field?

The methods are sound, and I have no concerns about them. However, as I said earlier in my comments, for a better estimate of the atmospheric fluxes, they should have done probably assessments of ebullition.

Is there enough detail provided in the methods for the work to be reproduced?

Yes, they is enough great information about the methods and I have no concerns about their reproducibility.

Reviewer #2 (Remarks to the Author):

Review of "Groundwater discharge as a drive of methane emissions from Arctic lakes" by Olid and colleagues

The authors of the paper conducted an important study on quantifying groundwater discharge and groundwater-delivered methane to a several lakes in the subarctic region of Sweden characterized by sporadic permafrost (that is permafrost is present but discontinuous). They unequivocally found that groundwater is entering the lakes and that this groundwater has high levels of methane, and is thus a major source of methane for the lake. The topic and the findings are timely and of broad importance.

The findings of the paper rest on the method used to estimate groundwater discharge. This was accomplished through using the tracer ^{222}Rn in a box model (mass balance) framework. This technique is widely used and certainly appropriate. The technique is predicated on quantifying all input/output and sink/source terms, with groundwater discharge flux as the unknown term to be calculated. Despite wide application, the technique can be fraught with uncertainty. The investigators handled uncertainty in a deterministic way, and in fact conservatively. They chose a ^{222}Rn concentration for groundwater that is relatively high (following a few measurements) which would give a lower estimate for groundwater inflow. This seems reasonable. However, a more robust uncertainty analysis would help the study. The approach the authors used has been wanting in terms of uncertainty quantification. Because of this, the authors should consider more analysis to make the study more robust. Related to this are improvements in terms of presentation of the raw data which are terms in the mass balance model. I offer more concrete recommendations and comments below. This work can go really far with some extra steps.

Good luck with further improving the study and the revision of the manuscript.

-Bayani Cardenas

This section focuses on uncertainty quantification. The mass balance (box) model is begging and ripe for a stochastic approach. It would be beneficial if the authors could pursue this, e.g., a Monte Carlo approach. I am happy to discuss how such analysis can be implemented.

GW endmember concentration (i.e., C_{gw} in the manuscript)- I could not tell how many samples were collected. Please provide this information. The model inputs (measurement results) and outputs would benefit from being presented as histograms or violin plots. The boxplots do not do justice and in fact can be misleading.

They only analyzed 10 mLs of porewater ample, albeit using liquid scintillation counting, rather than a RAD7. I am not an expert on this. However, I suspect that this is because they could not extract much groundwater. One wonders how heterogeneous the groundwater is in terms of radon concentration.

Table S1 and S2 shows a single value for C_{gw} . This seems to be assumed rather than measured. This must be the value chosen from the distribution. The proper way would have been to measure C_{gw} for each lake at each time. This certainly varies in space (potentially quite significantly) and through time. The assumption of a value and ensuing analysis is questionable without addressing this issue. It does not seem to make much sense when some other terms are lake specific,

whereas C-gw is not.

So how does one go around the issue of C-gw variability? One path forward is to do a Monte Carlo analysis guided by all C-gw values they measured. I would argue that they consider such an approach from end-to-end and across all the analysis (including for methane budget potentially).

There are other terms in the analysis that could be handled better. C-Ra is the same value. This is probably the detection limit for the instrument, i.e., it is "zero". I do not recall reading how this value was chosen; I may have missed it. In any case, it also ideally lake-specific.

The authors note interquartile ranges in the text and in the figures (boxplots). This is difficult when plotting in log-space. It is really more meaningful and transparent to show histograms. Please do so.

Some general and sporadic comments below:

Porewater sampling – it matters a lot whether they pulled from organic layers (i.e., peat in the active layer) or from mineral-rich horizons. While the latter will have high C_{gw}, they may not be the main source of gw discharge because of lower permeability. On the other hand, some organic layers have high permeability and thus discharge, but may have little radon because of lower production rates and residence times. The authors went around this by doing incubation experiments to quantify the max C_{gw}. This was confusing. All along, one gets the sense that porewater samples were used to constrain C_{gw}. The reality seems to be they were not used for anything. Nonetheless, it matters which sediments were used for the incubation. They noted that the incubation experiments were used to corroborate porewater measurements. I did not see any comparisons. I can only surmise that they used the incubation results for the mass balance analysis.

Please provide some sense of subsurface stratigraphy and permafrost distribution of the region, and ideally of the specific lakes.

The methane concentrations might also vary depending on what/where they are sampling. There can be very pronounced redox gradients in the subsurface. Show the histogram of the data please.

L30: I am more familiar with discontinuous permafrost, rather than sporadic

In relation to the above, some background on the subsurface stratigraphy/geology would help. They note the active layer numerous times, yet given that they are working in discontinuous permafrost, they are probably dealing with intra-permafrost gw flow, and not just supra-permafrost gw flow through the active layer. This adds tremendous uncertainty. Do the lakes tend to have taliks? How extensive are they, if present?

L51: Many colleagues who have spent their careers on CH₄ (and in general C cycling) in lakes, including in northern latitudes, would find this statement appalling.

L166: Landscape-scale? Perhaps use 'regional' instead. 'Landscape groundwater' does not make sense.

Can some parts of the lakes be losing? That is, the lakes might be recharging groundwater. The mass balance model seems to ignore that possibility. Lakes can be losing, gaining, or flow-through (combination) in relation to groundwater. They start with the assumption that the BD lakes are gaining. What would help in conceptualizing this would be topographic maps. Please present them. Actually, I would appreciate aerial images (i.e., Google Earth images) and photographs of the lakes and landscapes studied.

Fig S1. Please indicate in the map where gw/porewater samples were collected for both ²²²Rn and CH₄.

Reviewer #3 (Remarks to the Author):

General Comments:

This paper describes new evidence of the role that groundwater inputs play in the emission of CH₄ from lakes in high-latitude (sporadic) permafrost environments. This is an important but largely understudied topic, and as such, this paper provides a valuable contribution to the literature and to the broader community of readers interested in the impacts of warming on GHG emissions in high-latitude permafrost environments.

Indeed, the role of groundwater inputs to lakes in this scope has not been well characterized as few studies exist to my knowledge that have quantified groundwater discharge and CH₄ export to northern lakes in permafrost terrain. This study sets a high bar by (1) evaluating the importance of groundwater contributions to the CH₄ budget of northern lakes (as well as other sources, such as stream inputs and ebullitions), (2) comparing groundwater CH₄ inputs to the amount of CH₄ emitted from these lakes, (3) comparing a degree of spatial and temporal variability across the landscape, and (4) characterizing potential watershed features that drive spatial variability in groundwater CH₄ inputs.

The study appears to have been carried out carefully. The various aspects of the study's methods are well described (e.g., Rn mass balance calculations and assumptions), and the paper is overall well written, composed, and engaging to the reader.

A stronger description of the role of soil organic carbon could be included given its potential importance in driving CH₄ concentrations in groundwater entering lakes in high-latitude permafrost environments. If soil organic carbon was not explicitly measured in this study, are there other studies from the region where relevant data can be pulled from (e.g., soil organic carbon concentrations across the landscape and at depths relevant to groundwater flow during the summer and fall)? If not, are there local interpolated geospatial maps available that could enable this information to be included in the analysis? There are certainly some excellent global maps of high-latitude soil organic carbon content that could be used (e.g., databases in Hugelius et al., 2020; <https://www.pnas.org/content/117/34/20438>).

This discussion, and to some degree the introduction, could be improved with a stronger description of the role of soil organic carbon, including its interaction with active layer depth, subsurface groundwater flow, thawing permafrost (if relevant), and other landscape features discussed in the paper. If the data were available (either from field-collected measurements or geospatial maps), I would encourage the inclusion of this information in the study's analysis (e.g., PCA) and interpretation of the mechanisms and drivers of groundwater CH₄ inputs (which would also be relevant to other CH₄ sources to lakes).

Specific Comments:

Abstract

Line 28 – “thawing permafrost”

Line 36 – consider replacing “large role” with a more specific description

Introduction

I would like to see a bit more description of how CH₄ is generated and exported to lakes via groundwater inflow in these sporadic permafrost zones of Sweden. Are these pathways similar or different to those described by other studies in other regions of the world (e.g., Toolik in Alaska)?

Results and Discussion

Active layer depth is mentioned as an important variable for soil CH₄ generation resulting in high groundwater CH₄ concentrations, but the discussion around why this is the case could be strengthened. For example, by including more details about the interactions with potentially

organic carbon rich soils at the surface and organic carbon poor soils at the bottom of the active layer; shifts in soil type, grain size, or texture with depth; and changes in subsurface hydrology and water residence time with depth.

I found myself wondering whether other constituents relevant to biological CH₄ dynamics (e.g., dissolved organic carbon and nitrogen or dissolved ammonium) were measured in the sampled groundwater, streams, and lakes? If so, it would be interesting to fold this information into the results and discussion, which could potentially further support the authors interpretations of the mechanisms responsible for the high CH₄ in groundwater inputs, but comparably lower CH₄ emissions from the lakes.

Methods

It appears that a description of how the spatial variables described in the paper/PCA (catchment area, lake area, watershed slope and local precipitation) were collected is lacking, and should be included.

Figures

If space permits, a map as a main figure would be nice to see as opposed to only included it in the SI.

Reviewer #1

General comments:

This is very well-written manuscript (ms) and I did not have issues neither with its writing style nor with the structure. Additionally, all the collected data used in this ms are easy to access and verify. I commend the authors for presenting their work in this clean format that was easy to follow.

The authors have set ambitious goals to shed light on the main drivers of methane (CH₄) emissions from lakes in the Arctic. Hence, the ms presents the results from (1) seasonal regional scale assessments of CH₄ fluxes derived by groundwater discharge for 10 lakes in the Arctic, and (2) comparison to concurrently measured atmospheric fluxes.

We thank the reviewer for her/his positive and constructive comments.

Rev1.1: The idea that gw discharging to Arctic lakes is a source of GHG (specifically CH₄) is not new. Paytan et al. (2015) was the first work that demonstrated the magnitude of CH₄ fluxes from Lake Toolik (Alaska) using the same approaches. After revising the published research on the subject, I think that the somewhat novelty of this work is that it attempts to examine the drivers of CH₄ fluxes. This examination was done using concurrently collected environmental data and PCA analyzes (Figure 4). While these analyses could be useful with certain degree of confidence, but I found the authors' attempt to reveal the mechanisms that drive the changes in fluxes relatively weak.

Overall, I think this is great work that could be published in some other lower impact journal. My main concerns are that the work makes some progress in the right direction of bringing more attention to gw as a source of CH₄ to Arctic lakes, yet (1) this is not a new idea, and (2) the identified drivers (e.g. temperature) are somewhat already known/established in the field of high latitude wetland studies (e.g. Saarnio et al., 1997; Luascu et al, 2012 to mention just a couple).

Response: Thank you for your comment. We understand that our study goes an important step further than (Paytan et al., 2015) in the sense that it extends the study to a regional level, establishes a strong relationship between groundwater inputs and emissions among lakes, and also delves deeper into the mechanisms by which CH₄ inputs through groundwater may increase under a climate change. Thus, this new study focuses not only on results from one single lake that could have very special characteristics associated with the area studied and the time of the year. In addition, our study evaluates, for the first time, the spatial and temporal patterns for groundwater CH₄ inputs to several lakes in the discontinuous permafrost region in Sweden. Understanding these patterns is crucial for assessing the importance of groundwater discharge for lake CH₄ emissions at large scales. To highlight the role of groundwater inflows for lake CH₄ emissions, we have now put our results in a broader context, including the results from Paytan et al., (2015) and Dabrowski et al., (2020) with a literature survey of the major fluxes of CH₄ (diffusion, ebullition, production, and oxidation) in Arctic lakes (see Figure 4). This provides an important context and overview of lake CH₄ fluxes.

As mentioned above, this study not only explores into the mechanisms by which more CH₄ concentration may be generated, such as temperature, but also the mechanisms by which groundwater CH₄ input may increase under a climate change scenario. Groundwater CH₄ inputs depend not only on porewater CH₄ concentration in the mires, but also on the magnitude of the groundwater flow. Unlike CH₄ concentrations in mires, drivers of groundwater flows into lakes in the permafrost region are poorly understood, which are fundamentally different from those impacting in situ CH₄ production rates. For instance, our results show a negative relationship between peatland coverage and groundwater inflow rates, resulting in a limited supply of CH₄ despite the high production in these systems during summer. This spatial exploration of the drivers is also a novel aspect of this work in relation to the only two studies published up to date on this topic (Dabrowski et al., 2020; Paytan et al., 2015), since we explore the drivers of groundwater CH₄ inputs. This information is a necessary step forward to improve predictions of CH₄ emissions from Arctic lakes.

We have now emphasized that the novelty of our study lies in 1) assessing spatial and temporal patterns of groundwater discharge and CH₄ export to Arctic lakes in the sporadic permafrost region and 2) disclosing the major drivers of groundwater and CH₄ fluxes (Lines 23 – 35, 58 – 72).

Rev1.2: Does the work support the conclusions and claims, or is additional evidence needed? Some claims are better supported than others. For example, the significance of gw as a source of CH₄ is well supported by the data after being compared to atmospheric fluxes (with one caveat, they assessed only gas diffusion and did not assess ebullition). However, their analyzes on the drivers such as temperature, water depth etc., are not well-supported to my opinion. They made some assumption in their interpretations and suggest “more future research on the subject needs to be done”.

Response: Thank you for considering that the significance of groundwater as a source of CH₄ to lakes has been well supported even though no estimate of ebullition was addressed. In this new version of the manuscript, we have included this estimate to give robustness to the results obtained. To provide an indicative estimate of ebullition fluxes from the lakes, we used the results from 9 years of flux measurements in three lakes located in the Stordalen mire (Jansen et al., 2019). For lakes in Stordalen, diffusive CH₄ emission accounted for 17 – 52% of the ice-free CH₄ flux, with the remainder being emitted via ebullition. The Stordalen lakes are situated in a unique palsa mire complex rich in organic matter, which is not fully representative of this landscape, likely leading to ebullition contribution more to the total lake emissions compared to lakes in less organic settings. Hence, to estimate maximum ebullition and maximum total lake emission - thereby making our comparisons between groundwater inputs and total emissions conservative - we assumed that diffusive flux and ebullition accounted for 17% and 83%, respectively, in all our lakes. It should be noted that the ebullition data from the three lakes in Stordalen is of exceptional quality, with near 15,000 consistent measurements of ebullition over 9 years. The errors in extrapolating these numbers to similar nearby lakes are likely smaller than the errors from rather few scattered ebullition measurements affordable in normal lake CH₄ flux studies due to the large intralake variability (up to 74%) (Wik et al., 2013). Also, the range of estimated ebullitive CH₄ fluxes were well within the range reported for ebullitive fluxes from glacial and post glacial lakes across the boreal and arctic region (Line 128,). We are aware that these estimates only allow us to obtain a rough estimate of ebullitive CH₄ emissions. However, these results reveal that total CH₄ emissions are comparable in magnitude as groundwater CH₄ inputs (Figure 4), and thus support our main hypothesis that groundwater discharge could sustain total CH₄ emissions from the lakes (Lines 131 – 150).

In relation to the second comment, analyses on the drivers of groundwater discharge flow rates have been improved by performing a partial least squares (PLS) regression analysis. PLS is especially useful when predictor variables are correlated, and when there are more predictor variables than observations. Unlike PCA, PLS shows the discrimination between variables, which allowed us to evaluate which variables were most relevant for describing groundwater inflow variability. We combined the PLS with a multiple linear regression to better discuss the interaction between the different important variables (Lines 152 – 174).

Rev1.3: Are there any flaws in the data analysis, interpretation and conclusions? - Do these prohibit publication or require revision?

I don't find any flaws in the data analysis, but some of the interpretations and then statements are somewhat far reaching. For example, in order to demonstrate the magnitude of the gw-derived CH₄ fluxes, they compared them to atmospheric fluxes (which is great!). However, they only calculated the diffusive CH₄ fluxes to the atmosphere but neglected the ebullition (bubbles formed within the permafrost and the lake water column). On page 7-8 line 125-129 they concluded, that “While diffusive fluxes are often lower than ebullition and often account for 20-50% of the sum of the total CH₄ in similar lakes, their results suggest that gw inflow of CH₄ is comparable in magnitude to the CH₄ flux to the atmosphere by BOTH diffusive flux and ebullition.” I consider this is a relatively big assumption and find it a weakness in their interpretations.

Response: We have included estimations of ebullition fluxes to estimate total CH₄ emissions from the lakes in the new version of the manuscript to reinforce the statements of the text (see Response to Rev1.2).

Rev1.4: Similar statement is made on page 8 lines 130-144. At the beginning of the paragraph the authors state that their work “confirms” that gw is an overlooked source of CH₄ (hence I understand not a new idea), but their work is unique in a way that it is based on multiple lakes, i.e. it is an ubiquitous process. While I think taking their findings just in the context of high latitude lakes is perhaps true, Paytan et al. (2015) and Lecher et al (2015) demonstrated in two-year study that this process is indeed ubiquitous not only in lakes, but also across various permafrost coverage in marine and fresh water systems. I find the findings by Paytan’s group having more weight on the arguments of ubiquitous distribution. Hence, the current statement not extremely significant.

Response: We agree with the reviewer that “ubiquitous” was not the most convenient word to use here, as we are demonstrating the relevance of groundwater discharge as a source of CH₄ in several lakes in the discontinuous permafrost region but not in any other water system. We have modified this statement and clarified why our study differs from those by (Lecher et al., 2017 and Paytan et al., 2015) (see Response to Rev1.1) (Lines 25- 35). In order to focus our study more on lakes, we have now included a new figure with a literature survey of the main CH₄ fluxes in Arctic lakes, highlighting that groundwater inflows have the same magnitude as diffusion, ebullition or sediment methane production (Figure 4).

Rev1.5: Is the methodology sound? Does the work meet the expected standards in your field? The methods are sound, and I have no concerns about them. However, as I said earlier in my comments, for a better estimate of the atmospheric fluxes, they should have done probably assessments of ebullition.

Response: We have assessed ebullition from the study lakes and included these fluxes when assessing total CH₄ emissions (see Response to Rev1.2).

Rev1.6: Is there enough detail provided in the methods for the work to be reproduced? Yes, they is enough great information about the methods and I have no concerns about their reproducibility.

Response: Thanks.

Reviewer #2:

Review of “Groundwater discharge as a drive of methane emissions from Arctic lakes” by Olid and colleagues

The authors of the paper conducted an important study on quantifying groundwater discharge and groundwater-delivered methane to a several lakes in the subarctic region of Sweden characterized by sporadic permafrost (that is permafrost is present but discontinuous). They unequivocally found that groundwater is entering the lakes and that this groundwater has high levels of methane, and is thus a major source of methane for the lake. The topic and the findings are timely and of broad importance.

The findings of the paper rest on the method used to estimate groundwater discharge. This was accomplished through using the tracer 222-Rn in a box model (mass balance) framework. This technique is widely used and certainly appropriate. The technique is predicated on quantifying all input/output and sink/source terms, with groundwater discharge flux as the unknown term to be calculated. Despite wide application, the technique can be fraught with uncertainty. The investigators handled uncertainty in a deterministic way, and in fact conservatively. They chose a 222Rn concentration for groundwater that is relatively high (following a few measurements) which would give

a lower estimate for groundwater inflow. This seems reasonable. However, a more robust uncertainty analysis would help the study. The approach the authors used has been wanting in terms of uncertainty quantification. Because of this, the authors should consider more analysis to make the study more robust. Related to this are improvements in terms of presentation of the raw data which are terms in the mass balance model. I offer more concrete recommendations and comments below. This work can go really far with some extra steps.

*Good luck with further improving the study and the revision of the manuscript.
-Bayani Cardenas*

We welcome the positive and helpful comments made by the reviewer and the overall acknowledgment of this work. Below we respond to the individual comments of Reviewer #2.

Rev2.1: This section focuses on uncertainty quantification. The mass balance (box) model is begging and ripe for a stochastic approach. It would be beneficial if the authors could pursue this, e.g., a Monte Carlo approach. I am happy to discuss how such analysis can be implemented.

Response: We thank the reviewer for offering his help to implement this analysis. Groundwater and CH₄ fluxes have been reevaluated following the reviewer's suggestions. An explanation about how we did it has been included in Methods (Lines 298 – 310). Below we discuss some comments and considerations on the determination of groundwater flows and derived CH₄ inputs, which are not explicitly detailed in the manuscript, but may be worth for understanding the approach used:

Notice that the ²²²Rn flux (F_{gw} , Bq m⁻² d⁻¹) to each lake has been calculated with a simple steady-state mass balance, accounting all possible sources/sink terms of this radionuclide. Its uncertainty (ΔF_{gw} , Bq m⁻² d⁻¹) has been determined in a deterministic way following basic uncertainty propagation rules. Subsequently, F_{gw} and ΔF_{gw} have been used to generate random data using normal distributions. The water and CH₄ fluxes have been estimated by using these generated data and randomly selecting the ²²²Rn concentration in groundwater (C_{gw}). We have followed this approach for the following reasons:

- (1) One could argue that uncertainty of ²²²Rn flux estimation (F_{gw}) could be better constrained by using a Monte Carlo analysis (i.e., generating random data for each source/sink terms of the mass balance). However, in this study the greatest source of uncertainty in the calculations of groundwater and CH₄ fluxes is by far the concentration in the discharging groundwater (C_{gw}). Notice that ²²²Rn concentrations in groundwater vary one order of magnitude within the interquartile range whilst relative uncertainty of the ²²²Rn flux is only about 20%. Therefore, we believe that the determination of the ²²²Rn flux using a Monte Carlo analysis will not result in better estimates of groundwater discharge or its uncertainty.
- (2) Computing the advective inflow of groundwater has been done by randomly selecting the ²²²Rn concentration in groundwater (C_{gw}). Instead, this calculation could be performed by adjusting a theoretical distribution to the endmember data (Figure 1 below), to subsequently use the best fitted distribution for generating random data in a Monte Carlo-like approach. The latest was our first approach, yet after adjusting different theoretical distributions (e.g., normal, beta, gamma) by using least-square fitting, none of them resulted in a satisfactorily fitting ($R^2 \ll 0.6$). Hence, we believe that the approach finally chosen is more reliable than a Monte Carlo approach, which undoubtedly would have generated artefacts on the further groundwater flow estimates.

Finally, some improvements have been made on the determination of CH₄ fluxes. In contrast with previous calculations that were made treating independently the ²²²Rn and CH₄ concentrations in groundwater, the new approach is based on using the ratio C_{CH_4-gw}/C_{Rn-gw} at each sampling point to compute the fluxes. This may significantly reduce the variability of the resulting CH₄ fluxes.

Figure 1. Distribution of ^{222}Rn activities in groundwater samples ($n = 41$)

Rev2.2: GW endmember concentration (i.e., C_{gw} in the manuscript)- I could not tell how many samples were collected. Please provide this information. The model inputs (measurement results) and outputs would benefit from being presented as histograms or violin plots. The boxplots do not do justice and in fact can be misleading.

Response: We collected 41 groundwater samples from the lake shoreline (Supplementary Figure 9). To better constrain equilibrium ^{222}Rn concentrations in groundwater, we additionally carried out 28 sediment incubation experiments. We have clarified this information in the revised version of the manuscript (Line 374 – 378) and Supporting Information (section S.3). ^{222}Rn concentrations in groundwater estimated using both approaches are included in Supplementary Data 2. Finally, we have replaced the boxplots with violin plots, as suggested by the reviewer (Supplementary Figure 1 and 2).

Rev2.3: They only analyzed 10 mL of porewater sample, albeit using liquid scintillation counting, rather than a RAD7. I am not an expert on this. However, I suspect that this is because they could not extract much groundwater. One wonders how heterogeneous the groundwater is in terms of radon concentration.

Response: Instead of using a semiconductor detector (RAD7), groundwater samples were measured using low background liquid scintillation counting (Quantulus 1220). This latter technique allows accurately quantifying ^{222}Rn concentrations in low-volume samples (10 mL) and minimizing the water-air interaction during sample collection.

Concentrations of ^{222}Rn in groundwater varied up to a factor of 4, likely due to the high lateral and vertical variability of mineral content in peat (Morison et al., 2017; Pawlowski et al., 2014). Despite the high variability in groundwater ^{222}Rn concentrations, the observed variability was lower than for other lakes in permafrost regions (Dabrowski et al., 2020; Paytan et al., 2015). We have included a short discussion about the variability of groundwater in terms of ^{222}Rn concentration in the Supporting Information (Lines 130 – 137, Supplementary Figure 10).

Rev2.4: Table S1 and S2 shows a single value for C_{gw} . This seems to be assumed rather than measured. This must be the value chosen from the distribution. The proper way would have been to measure C_{gw} for each lake at each time. This certainly varies in space (potentially quite significantly) and through time. The assumption of a value and ensuing analysis is questionable without addressing

this issue. It does not seem to make much sense when some other terms are lake specific, whereas C-gw is not.

Response: We agree that the best way to constrain ^{222}Rn concentrations in groundwater (C_{gw}) is having a representative set of samples for each lake in both seasons. However, this requires the collection of several (> 10) groundwater samples from each lake to account for the expected spatial variability. Indeed, results of ^{222}Rn concentrations within the study lakes showed a variation of up to a factor of ~ 40 . Since all lakes were in the same area with very similar geological and topographical characteristics, it is expected that groundwater in the different lakes would have similar ^{222}Rn ranges than those found within the same lake. Thus, considering the available dataset, we believe it is more appropriate producing a unique representative C_{gw} that accounts for the variability found in all the lakes, rather than producing C_{gw} specific for each lake that might be highly biased towards the type of groundwater sampled (organic or more mineral-rich horizons). The landscape across lakes is quite homogeneous (Supplementary Figures 6, 7, 8, and 10), and thus this pooled sample likely captures the average groundwater content. Similarly, considering that the ^{226}Ra content present in the geological material is the main driver of ^{222}Rn concentrations in groundwater, C_{gw} is not expected to change significantly over time and we prefer to report a unique value for both seasons. This has been clarified in the Supporting Information.

Rev2.5: So how does one go around the issue of C-gw variability? One path forward is to do a Monte Carlo analysis guided by all C-gw values they measured. I would argue that they consider such an approach from end-to-end and across all the analysis (including for methane budget potentially).

Response: We have performed a Monte Carlo analysis using all C_{gw} values we measured (see Response to Rev2.1), as suggested by the reviewer.

Rev2.6: There are other terms in the analysis that could be handled better. C-Ra is the same value. This is probably the detection limit for the instrument, i.e., it is “zero”. I do not recall reading how this value was chosen; I may have missed it. In any case, it also ideally lake-specific.

Response: $C\text{-Ra}$ was measured in the waters of five of the study lakes. In three of them, $C\text{-Ra}$ was below the detection limit (average of 6 Bq kg^{-1}). This information is now included in the Supporting Information of the manuscript (Line 82). In those lakes where ^{226}Ra was detected in lake water, the concentration ranged from 11 to 21 Bq kg^{-1} . However, ^{222}Rn production by ^{226}Ra decay was a minor term in the ^{222}Rn mass balance (below 3% of the total ^{222}Rn inputs) (see Supplementary Figure 3). Therefore, we assumed negligible concentrations of ^{226}Ra content in lake waters where measurements were not available. We have added more detailed information about $C\text{-Ra}$ and its implications in the ^{222}Rn mass balance in the Supporting Information (Lines 79 – 86).

Rev2.7: The authors note interquartile ranges in the text and in the figures (boxplots). This is difficult when plotting in log-space. It is really more meaningful and transparent to show histograms. Please do so.

Response: According to the previous comments of the reviewer, we have replaced the box plots for groundwater concentrations with violin plots (Supplementary Figures 1 and 2). We have also added a histogram in the Supporting Information to show the values obtained for the groundwater ^{222}Rn concentration using different approaches (i.e., direct groundwater measurements and sediment incubations) (Supplementary Figure 9). This histogram does not only allow to evaluate the high spatial variability of the groundwater ^{222}Rn concentration addressed already in a previous comment, but also to compare the two methodologies used: groundwater sample collection and equilibration experiments.

Some general and sporadic comments below:

Rev2.8: Porewater sampling – it matters a lot whether they pulled from organic layers (i.e., peat in the active layer) or from mineral-rich horizons. While the latter will have high C_{gw} , they may not be the

main source of gw discharge because of lower permeability. On the other hand, some organic layers have high permeability and thus discharge, but may have little radon because of lower production rates and residence times. The authors went around this by doing incubation experiments to quantify the max C_{gw} . This was confusing. All along, one gets the sense that porewater samples were used to constrain C_{gw} . The reality seems to be they were not used for anything. Nonetheless, it matters which sediments were used for the incubation. They noted that the incubation experiments were used to corroborate porewater measurements. I did not see any comparisons. I can only surmise that they used the incubation results for the mass balance analysis.

Response: We agree that the discussion of the groundwater endmember was confusing. We have now clarified the points raised by the reviewer in the new version of the manuscript (Lines 348 – 357, 374 – 378). Measurements in porewater samples, which were mainly collected from organic layers right at the lake shoreline, were used to determine C_{gw} . Additionally, we conducted sediment incubation experiments in the lab to better characterize this parameter. Results from incubation experiments are only used to validate the ranges of concentrations measured with an independent estimate, but they are not used in the calculations. As suggested, we have now included a comparison of the results from both methods, and we have clarified this issue in the Supporting Information (see section S.3, Supplementary Figure 10).

Rev2.9: Please provide some sense of subsurface stratigraphy and permafrost distribution of the region, and ideally of the specific lakes.

Response: Unfortunately, we are unable to include this information in the manuscript as we do not have any information about subsurface stratigraphy of the area. We however have now included a map showing the permafrost distribution in the region and for the study lakes (Supplementary Figure 6).

Rev2.10: The methane concentrations might also vary depending on what/where they are sampling. There can be very pronounced redox gradients in the subsurface. Show the histogram of the data please.

Response: We now provide a violin plot to compare the concentration of CH_4 found in the different sources (i.e., groundwater, lake water, inlet and outlet streams) (Supplementary Figure 2). This graph provides information about the distribution of CH_4 concentration in each water source. We think that providing a histogram of CH_4 concentrations in groundwater would be thus redundant. We however will include it in case the review thinks it is still needed.

Rev2.11: L30: I am more familiar with discontinuous permafrost, rather than sporadic

Response: Discontinuous permafrost is broken up into separate regions. Discontinuous permafrost can be isolated or sporadic. It is called isolated if less than 10% of the surface has permafrost under it. Sporadic means 10 to 50% of the surface has permafrost under it. In the Abisko area, the occurrence of continuous permafrost is around 6%, and discontinuous permafrost make up 35% of the area, while the remaining 59% is sporadic (Hällberg, 2018). However, we agree with the reviewer that maybe a more general term here is valid. So, we have replaced the word “sporadic” by “discontinuous” in the text.

Rev2.12: In relation to the above, some background on the subsurface stratigraphy/geology would help. They note the active layer numerous times, yet given that they are working in discontinuous permafrost, they are probably dealing with intra-permafrost gw flow, and not just supra-permafrost gw flow through the active layer. This adds tremendous uncertainty. Do the lakes tend to have taliks? How extensive are they, if present?

Response: Unfortunately, we do not have detailed information for those small catchments about the stratigraphy and geology of the area. Permafrost occurrence though is low, often restricted to mires and higher elevation areas (Johansson et al., 2006). To partially assess this, we have included a topographic and land cover map (Supplementary Figure 7). Due to the use of only one isotope (^{222}Rn), we are not able to distinguish between different sources of groundwater (i.e., intra- and supra-permafrost

groundwater). For that reason, we use the term “groundwater” to refer to any type of water that circulates below the surface.

Rev2.13: L51: Many colleagues who have spent their careers on CH₄ (and in general C cycling) in lakes, including in northern latitudes, would find this statement appalling.

Response: We have re-written this statement. The text now says: “*Yet the sensitivity of CH₄ emissions from Arctic lakes to climate change is highly uncertain because of poor understanding of the underlying mechanisms controlling lake CH₄ cycling*” (Lines 43 – 46).

Rev2.14: L166: Landscape-scale? Perhaps use ‘regional’ instead. ‘Landscape groundwater’ does not make sense.

Response: We have modified the text according to the reviewer’s comment.

Rev2.15: Can some parts of the lakes be losing? That is, the lakes might be recharging groundwater. The mass balance model seems to ignore that possibility. Lakes can be losing, gaining, of flow-through (combination) in relation to groundwater. They start with the assumption that the BD lakes are gaining. What would help in conceptualizing this would be topographic maps. Please present them. Actually, I would appreciate aerial images (i.e., Google Earth images) and photographs of the lakes and landscapes studied.

Response: We agree that is possible that some lakes are losing groundwater. This is however really hard to assess and, given that all of our lakes are located in the lowest elevations of the landscape, this is unlikely. We have now clarified this in the document (Lines 280 – 281), stating that we take a conservative approach and assume that the lakes are not losing groundwater, i.e. we are providing the minimum amount of groundwater discharging into the lakes. We have included a topographic and land cover map (Supplementary Figure 7).

Rev2.16: Fig S1. Please indicate in the map where gw/porewater samples were collected for both 222Rn and CH₄.

Response: We have provided the location of the groundwater sampling points (Supplementary Figure 9).

Reviewer #3:

General Comments:

This paper describes new evidence of the role that groundwater inputs play in the emission of CH₄ from lakes in high-latitude (sporadic) permafrost environments. This is an important but largely understudied topic, and as such, this paper provides a valuable contribution to the literature and to the broader community of readers interested in the impacts of warming on GHG emissions in high-latitude permafrost environments.

Indeed, the role of groundwater inputs to lakes in this scope has not been well characterized as few studies exist to my knowledge that have quantified groundwater discharge and CH₄ export to northern lakes in permafrost terrain. This study sets a high bar by (1) evaluating the importance of groundwater contributions to the CH₄ budget of northern lakes (as well as other sources, such as stream inputs and ebullitions), (2) comparing groundwater CH₄ inputs to the amount of CH₄ emitted from these lakes, (3) comparing a degree of spatial and temporal variability across the landscape, and (4) characterizing potential watershed features that drive spatial variability in groundwater CH₄ inputs.

The study appears to have been carried out carefully. The various aspects of the study’s methods are

well described (e.g., Rn mass balance calculations and assumptions), and the paper is overall well written, composed, and engaging to the reader.

We thank the reviewer for her/his positive comments about the manuscript.

Rev3.1: A stronger description of the role of soil organic carbon could be included given its potential importance in driving CH₄ concentrations in groundwater entering lakes in high-latitude permafrost environments. If soil organic carbon was not explicitly measured in this study, are there other studies from the region where relevant data can be pulled from (e.g., soil organic carbon concentrations across the landscape and at depths relevant to groundwater flow during the summer and fall)? If not, are there local interpolated geospatial maps available that could enable this information to be included in the analysis? There are certainly some excellent global maps of high-latitude soil organic carbon content that could be used (e.g., databases in Hugelius et al., 2020; <https://www.pnas.org/content/117/34/20438>).

Response: Dissolved organic carbon (DOC) was measured in lake water and groundwater samples. DOC in the lakes was low and similar among lakes. An initial exploration of the data did not suggest a pattern with DOC and groundwater inflows, and thus was not further included in the analysis. We have now included original sources with descriptive characteristics of all lakes, including DOC concentrations table (see Supplementary Data 2).

We agree with the reviewer that exploring the role of SOC in the landscape would be of great value to understand groundwater inflows of CH₄ into lakes. We explored global gridded products as suggested by the reviewer, but they are not at a fine resolution enough to be meaningful in this study. For example, the work by (Hugelius et al., 2020) has a pixel size of 10 km (area of 100 km²). In this particular mountain landscape, this product also predicts higher SOC at higher elevations, which is contrary to observations (Siewert, 2018). In a small catchment nearby, Stordalen, exists a study mapping SOC at high spatial resolution (Siewert, 2018). This work shows a fine scale variation in SOC, linked to topographic controls and especially high stocks in mires. In our study, we include landscape drivers such as percentage of mire cover and percentage of wet areas which likely capture those areas with larger carbon stocks in soils.

Rev3.2: This discussion, and to some degree the introduction, could be improved with a stronger description of the role of soil organic carbon, including its interaction with active layer depth, subsurface groundwater flow, thawing permafrost (if relevant), and other landscape features discussed in the paper. If the data were available (either from field-collected measurements or geospatial maps), I would encourage the inclusion of this information in the study's analysis (e.g., PCA) and interpretation of the mechanisms and drivers of groundwater CH₄ inputs (which would also be relevant to other CH₄ sources to lakes).

Response: We agree with the reviewer that SOC could interact importantly with groundwater inflows for lake biogeochemistry. In the discussion now we have included what effects we expect from climate change, among them the interaction of increased precipitation and permafrost thaw, with the subsequent C mobilisation (lines 216 - 230). As stated above, we cannot include SOC directly in the analysis due to the lack of data, but part of the patterns on SOC in this region are likely captured in other spatial predictors such as percentage of mire cover.

Specific Comments:

Abstract

Rev3.3: Line 28 – “thawing permafrost”

Response: The word has been replaced.

Rev3.4: Line 36 – consider replacing “large role” with a more specific description

Response: The abstract has been modified to specify more clearly the findings of this study.

Introduction

Rev3.5: I would liked to see a bit more description of how CH₄ is generated and exported to lakes via groundwater inflow in these sporadic permafrost zones of Sweden. Are these pathways similar or different to those described by other studies in other regions of the world (e.g., Toolik in Alaska)?

Response: We have now included a discussion about the differences in the magnitude of groundwater-derived CH₄ export via groundwater between our study sites and lakes in Alaska (Lines 103 - 110).

Results and Discussion

Rev3.6: Active layer depth is mentioned as an important variable for soil CH₄ generation resulting in high groundwater CH₄ concentrations, but the discussion around why this is the case could be strengthened. For example, by including more details about the interactions with potentially organic carbon rich soils at the surface and organic carbon poor soils at the bottom of the active layer; shifts in soil type, grain size, or texture with depth; and changes in subsurface hydrology and water residence time with depth.

Response: As we mentioned before (see Response to Rev3.1 and 3.2), the low spatial resolution of the SOC data available does not allow us to evaluate in detail the role of organic C content on controlling groundwater CH₄ concentrations. To do this, we would need SOC at a resolution of meters. Similarly, we do not have high spatial resolution data for soil type, grain size or texture with depth. We have now been able to provide spatial information on land cover, which has been included in the data analyses and discussion of the results. As we do not have these data, we do not think we should discuss the potential implications of all these factors on controlling groundwater discharge rates into the lakes.

Rev3.7: I found myself wondering whether other constituents relevant to biological CH₄ dynamics (e.g., dissolved organic carbon and nitrogen or dissolved ammonium) were measured in the sampled groundwater, streams, and lakes? If so, it would be interesting to fold this information into the results and discussion, which could potentially further support the authors interpretations of the mechanisms responsible for the high CH₄ in groundwater inputs, but comparably lower CH₄ emissions from the lakes.

Response: We agree with the reviewer that, besides directly transporting CH₄ to the lakes, groundwater may prompt CH₄ emissions from lakes by delivering organic carbon that can be mineralized to CO₂ and CH₄. We have added a brief discussion about this in the revised version of the manuscript (Lines 216 – 240). Besides DOC mentioned above, we do not have nutrient concentrations available to include in the study.

Methods

Rev3.8: It appears that a description of how the spatial variables described in the paper/PCA (catchment area, lake area, watershed slope and local precipitation) were collected is lacking, and should be included.

Response: A description of how the spatial variables were estimated has been included in the revised version of the manuscript (Lines 402 – 419).

Figures

Rev3.9: If space permits, a map as a main figure would be nice to see as opposed to only included it in the SI.

Response: The map has been included as a main figure (Figure 1).

REFERENCES

- Dabrowski, J. S., Charette, M. A., Mann, P. J., Ludwig, S. M., Natali, S. M., Holmes, R. M., et al. (2020). Using radon to quantify groundwater discharge and methane fluxes to a shallow, tundra lake on the Yukon-Kuskokwim Delta, Alaska. *Biogeochemistry*, *148*, 69–89.
- Hällberg, P. (2018). *Permafrost Modelling and Climate Change Simulations in Northern Sweden*. Uppsala University.
- Hugelius, G., Loisel, J., Chadburn, S., Jackson, R. B., Jones, M., MacDonald, G., et al. (2020). Large stocks of peatland carbon and nitrogen are vulnerable to permafrost thaw. *Proceedings of the National Academy of Sciences*, *117*, 20438–20446.
- Jansen, J., Thornton, B. F., Jarnet, M., Wik, M., Cortés, A., Friborg, T., et al. (2019). Climate-sensitive controls on large spring emissions of CH₄ and CO₂ from northern lakes. *Journal of Geophysical Research G: Biogeosciences*, *124*, 2379–2399. <https://doi.org/10.1029/2019JG005094>
- Johansson, M., Christensen, T. R., Akerman, H. J., & Callaghan, T. V. (2006). What determines the current presence or absence of permafrost in the Torneträsk region, a sub-arctic landscape in Northern Sweden? *Ambio*, *35*(4), 1–9. [https://doi.org/10.1579/0044-7447\(2006\)35\[190:WDTCP0\]2.0.CO;2](https://doi.org/10.1579/0044-7447(2006)35[190:WDTCP0]2.0.CO;2)
- Lecher, A. L., Chuang, P., Singleton, M., & Paytan, A. (2017). Sources of methane to an Arctic lake in Alaska: An isotopic investigation. *Journal of Geophysical Research: Biogeosciences*, *122*(4), 753–766. <https://doi.org/10.1002/2016JG003491>
- Morison, M. Q., Macrae, M. L., Petrone, R. M., & Fishback, L. (2017). Seasonal dynamics in shallow freshwater pond-peatland hydrochemical interactions in a subarctic permafrost environment. *Hydrological Processes*, *15*(2), 462–475. <https://doi.org/10.1002/hyp.11043>
- Pawlowski, D., Okupny, D., Wlodarski, W., & Zielinski, T. (2014). Spatial variability of selected physicochemical parameters within peat deposits in small valley mire: a geostatistical approach. *Geologos*, *20*. <https://doi.org/10.2478/logos-2014-0020>
- Paytan, A., Lecher, A. L., Dimova, N., Sparrow, K. J., Kodovska, F. G.-T., Murray, J., et al. (2015). Methane transport from the active layer to lakes in the Arctic using Toolik Lake, Alaska, as a case study. *Proceedings of the National Academy of Sciences*, *112*(12), 201417392. <https://doi.org/10.1073/pnas.1417392112>
- Siewert, M. B. (2018). High-resolution digital mapping of soil organic carbon in permafrost terrain using machine learning: a case study in a sub-Arctic peatland environment. *Biogeosciences*, *15*, 1663–1682. <https://doi.org/10.5194/bg-15-1663-2018>
- Wik, M., Crill, P. M., Varner, R. K., & Bastviken, D. (2013). Multiyear measurements of ebullitive methane flux from three subarctic lakes. *Journal of Geophysical Research: Biogeosciences*, *118*(3), 1307–1321. <https://doi.org/10.1002/jgrg.20103>

REVIEWERS' COMMENTS

Reviewer #2 (Remarks to the Author):

I reviewed the original version of this manuscript and will not repeat my previous comments. The authors have done an important investigation with some apparent loose ends from their previous analysis and from their original presentation. Those have all been thoroughly addressed. The manuscript is strong and its significant conclusions are defensible.

This paper is more or less ready to go. I spotted a few typographical and grammatical errors here and there. The authors need to do another very thorough editorial scrub.

Reviewer #3 (Remarks to the Author):

The authors seemed to have carefully considered the reviewer comments, and have done a thorough job revising their manuscript accordingly. The revisions have strengthened the paper in areas that were suggested. I do not have any additional major or minor comments to provide. The paper, in my opinion, is ready for potential publication in Nature Communications.

RESPONSE TO REVIEWERS

Reviewer #2

I reviewed the original version of this manuscript and will not repeat my previous comments. The authors have done an important investigation with some apparent loose ends from their previous analysis and from their original presentation. Those have all been thoroughly addressed. The manuscript is strong and its significant conclusions are defensible.

This paper is more or less ready to go. I spotted a few typographical and grammatical errors here and there. The authors need to do another very thorough editorial scrub.

We thank the reviewer for her/his positive comments. We have edited the text to correct some typographical and grammatical errors.

Reviewer #3

The authors seemed to have carefully considered the reviewer comments, and have done a thorough job revising their manuscript accordingly. The revisions have strengthened the paper in areas that were suggested. I do not have any additional major or minor comments to provide. The paper, in my opinion, is ready for potential publication in Nature Communications.

We welcome the positive comments made by the reviewer and the overall acknowledge of our work.